# The GravyTrain toolbox for molecular cell biology

Oren Shatz, Zvulun Elazar ⓘ*

Department of Biomolecular Sciences, The Weizmann Institute of Science, Rehovot, Israel

* zvulun.elazar@weizmann.ac.il

## Abstract

Yeast genetics has the power for thorough investigation of complex systems in molecular cell biology. Here, we present GravyTrain, a comprehensive repository of constructs for genomic modifications, including gene deletions or tagging with fusion tags for robust protein characterization and manipulation. The modular cloning scheme employed by GravyTrain allows shuffling of elements between constructs and potentially the application of included protein tags to research beyond yeast. The experimental potential of GravyTrain is demonstrated by the *de novo* construction of a library of strains for studying autophagy in yeast.

## Introduction

Budding yeast (or simply yeast hereafter) are a powerful model organism for studying molecular cell biology, offering a simple, fully sequenced haploid genome that is easily manipulated to investigate processes conserved in human [1]. The pYM plasmid library [2] allows easy genomic modification to any locus-of-choice by amplifying DNA pYM cassettes with universal S1/S2/S3/S4 adapters, flanked by genomic homology sequences. More specifically, proteins may be tagged by fusion proteins such as the red fluorescent protein DsRed [3], affinity tags such as TAP [4] and other useful tags. However, cloning of new constructs based on pre-existing publicly available pYM plasmids has so far been limited to restriction enzyme-based approaches. Alternatively, the SWAP-Tag approach allows seamless tagging and a modular cloning scheme based on the pYM adaptors, based on the specific S288c genetic background with a more limited collection of donor tags constructs [5], whereas the more recently developed CRISPR/Cas9-based tagging allows seamless tagging of any locus, yet requires an intermediate marker replacement step for genome editing to avoid locus-specific sgRNA cloning [6] .

Here we introduce GravyTrain, a toolbox that allows pYM-based genomic modification in yeast while supplying new genetic elements, including a diverse collection of protein tags, inducible promoters, synthetic terminators, and exogenous selection markers. The toolbox is built upon a restriction-free cloning scheme where elements are easily shuffled between different constructs, while largely avoiding lengthy

**Data availability statement:** All relevant data are within the manuscript and its Supporting Information files.

**Funding:** Israel Science Foundation (Grants 215/19 and 1272/24), and the Yeda-Sela Center for Basic Research.

**Competing interests:** The authors have declared that no competing interests exist.

homology regions between either GravyTrain constructs or endogenous genomic loci. Moreover, the GravyTrain community framework for protein tag shuffling bears the potential to apply tags to different model organisms and construct types. We demonstrate the applicability of the toolbox to future studies by the facile *de novo* construction of a comprehensive strain library for mechanistic molecular characterization of autophagy in yeast. Furthermore, we discuss the unique contribution of GravyTrain to the employment of a subset of these strains in the context of our recent yeast study of the roles of the rim of autophagic membranes. In conclusion, GravyTrain provides a methodological advance in the harnessing of easy and routine laboratory protocols of molecular cloning and yeast genetics to the potential of comprehensive studies in molecular cell biology.

## Results

### GravyTrain is a unified design for cloning of genetic elements in construct libraries

To provide a robust framework for homologous recombination-driven genome editing in yeast, we devised a set of constructs of different types (see sections below) with a common layout (Fig 1A (top), S1A Fig). In this layout all biologically active genetic elements, namely selection marker, promoter, terminator, protein linker and protein tags, are separated by restriction-free (RF) hotspots. Hotspots (S1 Table, sheet "Elements Details" under the "RF Hotspot" type) are short, unique DNA sequences with an approximate melting temperature of ~60C, applied in GravyTrain for creation of mega-primers in RF cloning [7]. By flanking all elements of a certain type (S1 Table, sheet "Elements Details") by the same hotspot pair, a universal primer design can be used to amplify this element type for restriction-free import of external elements into GravyTrain (S1B Fig), export from GravyTrain for external uses (S1C Fig) or shuffling across different GravyTrain constructs (Fig 1A (bottom), S1D Fig).

GravyTrain cassettes for yeast genetics are all listed (S2 Table, sheet "Construct Summary"). Canonical cassettes follow the pFA6a backbone and pYM standard for amplification, transformation and homologous recombination-based genomic integration [2], namely S1-S2 for knockout, S3-S2 for 3' tagging (e.g., for C-terminal protein tagging) and S1-S4 for 5' tagging (e.g., for promoter replacement and optional N-terminal protein tagging) – as well as S1-S4-S2 for knockin of exogenous expression – as detailed hereafter. The region between 5_A and A_M is currently empty in all included canonical cassettes, and is reserved for custom use, e.g., inclusion of additional genetic elements that are useful when genomically integrated together with the construct.

### GravyTrain tags for comprehensive protein characterization or manipulation

To combine the ease of the ubiquitously used standard library for protein tagging, such as the pYM [2], with the capability of previously-excluded or newly-developed tags or otherwise functional domains, GravyTrain introduces a comprehensive collection of tags for protein characterization or manipulation – applicable for C-terminal fusion by S3/S2 tagging or N-terminal fusion by S1/S4 genomic tagging

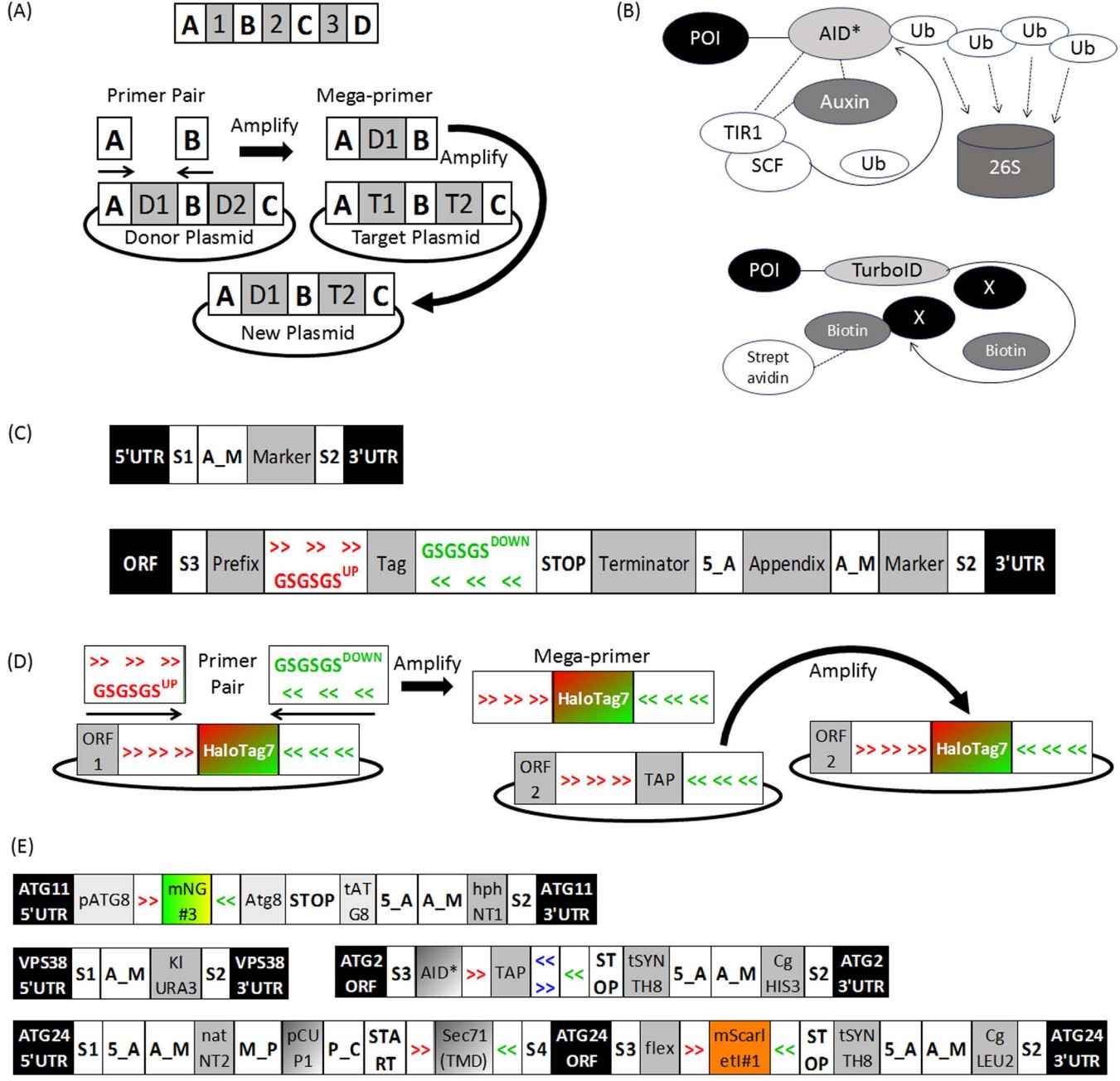

**Fig 1. The GravyTrain toolbox for molecular cell biology.** (A) (top) Common layout of GravyTrain constructs. Gray background – GravyTrain genetic elements. White background – a pair of GravyTrain restriction-free hotspots flank each genetic element. (bottom) The modular design of GravyTrain allows restriction-free shuffling of genetic elements between constructs. GravyTrain constructs contain genetic elements (gray background) flanked by restriction free hotspots for cloning (white background). A forward primer consisting of hotspot A sequence, and a reverse primer consisting of hotspot B reverse sequence, are first used to amplify the element D1 from a donor GravyTrain plasmid (containing elements D1, D2) into a mega-primer. Note that primer design is independent of shuffled element and both donor and target plasmids. Then, the mega-primer is used to amplify the target GravyTrain plasmid (containing elements T1, T2), thereby generating a new GravyTrain plasmid, wherein the original target element T1 has been substituted by the donated element D1. (B) GravyTrain provides diverse and powerful fusion tags for protein characterization or manipulation, e.g., AID* for auxin-induced, transient, and quantitative SCF$^{TIR1}$-mediated poly-ubiquitination and consequent degradation by the 26S proteasome (top), or TurboID for proximity-based labeling (bottom). (C) Canonical GravyTrain constructs for genome editing in yeast include, for example, the canonical pFA6a-S1-S2 for gene knockout (top) or pFA6a-S3-S2 constructs for 3' gene tagging (bottom), which are amplified by a reverse antisense S2 primer and forward sense S1 or S3 primer, respectively. The amplicon is then integrated between endogenous ORF and 3'UTR, thereby erasing the open reading frame

(ORF) of the protein (S1-S2) or tagging it (S3-S2) at its C-terminus with the Prefix-Tag moiety and substituting endogenous terminator function with the GravyTrain terminator included within the construct. (D) GravyTrain community constructs for broader applications of GravyTrain protein tags. To shuffle tags between different construct types where a different ORF is fused to the tag, the universal GSGSGS_UP (red forward arrows) and GSGSGS_DOWN (green backward arrows) primer pair are used to amplify mega-primers of a GravyTrain tag from a donor construct of one type and target them by another amplification to a construct of another type. This generates a new GravyTrain community construct of one type, where the target tag has been substituted by the donor tag from another construct type. (E) Instrumentation of GravyTrain for studying autophagy in yeast. Presented is a schematic representation of the GravyTrain elements within the ORS3280 strain genotype, including: integration of GravyTrain community ATG8 construct mNeonGreen-Atg8 for visualization of autophagic membranes, instead of the ORF of endogenous ATG11 to eliminate **Atg11**-mediated selective autophagy; C-terminal tagging of endogenous ATG2 ORF with GravyTrain construct of AID* for transient partial auxin-mediated depletion of **Atg2**. C-terminal tagging of endogenous **ATG24** ORF with GravyTrain construct of mScarletI for visualization of **Atg24**; N-terminal tagging of endogenous **ATG24** ORF with GravyTrain construct of TMD$^{Sec71}$ for tethering of **Atg24** to the cytoplasm leaflet of the ER membrane; knockout of VPS38 ORF for elimination of endosomal PI(3)P.

of endogenous yeast proteins of interest (POI). These fusion moieties are listed in a schematic fashion (Fig 2A, S1 Table, sheet "Elements Overview") and detailed (Fig 2B, S1 Table, sheet "Elements Details"), and can be classified with examples as follows (S2 Fig).

**Linker** – separation of POI from fused tags, e.g., flex (S2A Fig), a long flexible linker, derived from human carbonic anhydrase, originally used for fusing light and heavy antibody chains for a functional scFv polypeptide [8]. **Affinity** – affinity-based purification or detection of POI, e.g., our newly-introduced TAP (S2B Fig), comprising linker-separated (underlined) hexa-histidine peptide (blue) that binds nickel [9], two tandem copies of the 3xFLAG peptide (green) that binds M2 monoclonal antibody [10], and TwinStrepTag (orange) that binds StrepTactin and StrepTactin XT (IBA) [11]. **Degradation** – post-translational degradation of POI, e.g., AID* (Fig 1B (top), S2C Fig) for poly-ubiquitination and proteasomal degradation upon addition of exogenous auxin to the growth medium [12] or ubc9ts (S2D Fig) for constitutive, temperature-inflicted misfolding and proteasomal degradation [13]. **Visualization** – allow subcellular localization of POI by fluorescence microscopy, e.g., by the cyan-fluorescent superfolder mTurquoise2 (sfTq2) (S2E Fig) [14], or the chemically-tagged HaloTag7 [15] that supports conjugation to chemical dyes of various fluorescent properties, e.g., the Janelia JF series [16] (S2F Fig). **Targeting** – targeting of POI to a subcellular compartment, e.g., plasma membrane for Gpa2(PM) (S2G Fig) [17], molecular determinant, e.g., PI(3)P for the PX domain of Vam7 (S2H Fig) [18], or another tagged POI, e.g., HA_Nb (S2I Fig) [19] for non-covalent binding to another POI fused to HA, or SpyCatcher002 [20] for covalent binding to another POI fused to SpyTag002 (S2J Fig). **Modification** – moieties that change the properties of the fused POI, e.g., AaLS (S2K Fig) [21] that confers higher-order condensation on a 60-subunit nanoparticle surface, or of other proximal proteins, e.g., TurboID (Fig 1B (bottom), S2L Fig) [22] for biotin proximity labeling-mediated interactome studies.

Most GravyTrain constructs (e.g. pOS342) contain a single tag, flanked by upstream RF hotspot GSGSGS_UP and downstream RF hotspot GSGSGS_DOWN. However, GravyTrain also allows combination of different tags within a single polypeptide in two ways. First, in addition to the canonical tag element in constructs, another element type is the linker, which not only includes flex – which is solely a linker between domains – but also the function auxin-induced degron AID*. A combination of AID* with a canonical tag, e.g., TAP affinity tag (pOS343) or a fluorescent protein, e.g., sfTq2 (pOS467), allows both auxin-induced degradation of a protein, combined with affinity-based detection or microscopy-assisted visualization, respectively, of intact protein molecules. In addition, in some constructs, an additional **pivot** RF hotspot – GGSGGG – is inserted between the first and second tags in sequence. For example, pOS346 (S2 Table) is composed of GSGSGS_UP, TAP, GGSGGG, GSGSGS_UP, HaloTag7, GSGSGS_DOWN. However, we note that cloning tags flanked by GGSGGG rather than GSGSGS_UP/GSGSGS_DOWN require dedicated primers. Moreover, RF cloning of constructs including more than one pivot element (e.g. pOS353) is challenging and best avoided.

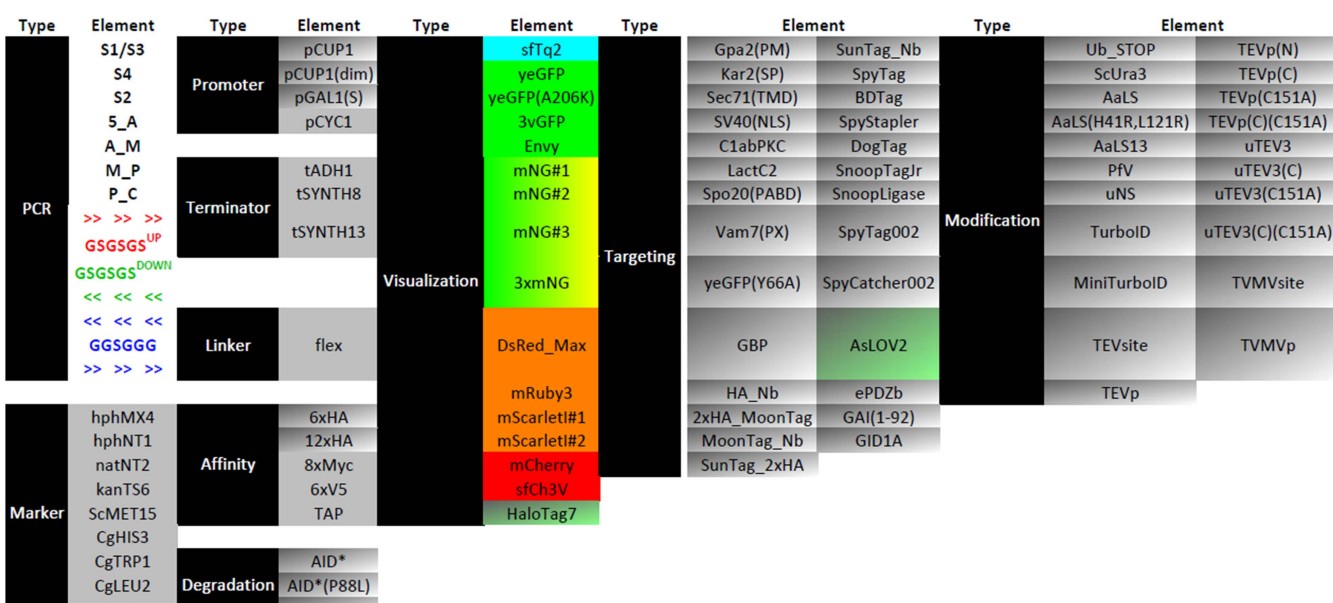

**Fig 2. Genetic Elements of GravyTrain.** (A) Overview of element types and names. (B) Exemplary description, amino acid sequence and DNA sequence for elements.

## Canonical pFA6a-S1-S2 constructs for gene knockout

Constructs for gene knockout are listed in S2 Table, sheet "pFA6a-S1-S2 constructs" and follow the standard pYM S1/S2 amplification scheme [2], comprising a common standard layout (Fig 1C (top), S3A Fig). In addition to previously published pYM natNT2 and hphNT1 antibiotic resistance [2], GravyTrain offers five heterologous auxotrophic markers for gene knockout – namely His, Met, Leu, Trp, or Ura – expressed under their native regulatory elements.

## Canonical pFA6a-S3-S2 constructs for 3' gene tagging

Constructs for 3' gene tagging are listed in S2 Table, sheet "pFA6a-S3-S2 constructs", and follow the standard pYM S3/S2 amplification scheme [2], comprising a common standard layout (Fig 1C (bottom), S3B Fig). The POI coded by the 3' tagged gene is fused at its C-terminus to one or more consecutive protein tags of the GravyTrain collection, followed by a STOP codon. In addition to the standard hphMX4 selection marker, GravyTrain offers five auxotrophic markers (as stated

above), of which mostly His and Trp are available for selection in 3' tagging cassettes. The STOP codon is directly followed by the strong constitutive, physiologically inert synthetic tSYNTH8 [23].

### Canonical pFA6a-S1-S4 constructs for 5' gene tagging

Constructs for 5' gene tagging are listed in S2 Table, sheet "pFA6a-S1-S4 constructs", and follow the standard pYM S1/S4 amplification scheme [2], comprising a common standard layout (S3C Fig), and are selected by Nat using the natNT2 marker as in the original pYM library [2]. The POI coded by the 5' tagged gene is fused at its N-terminus to one or more consecutive protein tags of the GravyTrain collection, followed by a START codon immediately prior to the protein. 5' tagging replaces the gene's original promoter with the GravyTrain cassette promoter, which for most cassettes is the copper-inducible pCUP1, supporting both basal and overexpression as previously shown [2]. Of note, GravyTrain introduces the pCUP1(dim) promoter (e.g., construct pOS268) that is tightly repressed and only fairly induced by copper [24], allowing a quantitative downregulation of protein expression at the transcript level [25].

### Canonical pFA6a-S1-S4-S2 constructs for knock-in of exogenous expression

In addition to genomic tagging or deletion, we combine the S1/S2 scheme for gene knockout and the S1/S4 scheme for pCUP1-driven protein expression to create chimeric copper-induced exogenous expression cassettes for genomic knock-in integration. These cassettes are listed in S2 Table, sheet "pFA6a-S1-S4-S2 constructs", and follow the standard pYM S1/S2 amplification scheme [2], comprising a common standard layout (S3D Fig) and are selected by Nat using the natNT2 marker, as S1-S4 constructs above. The S4 element of S1/S4 cassettes is here followed by a STOP codon and the mild synthetic, physiologically-inert terminator tSYNTH13 [23], enabling gene-independent, copper-driven expression of one or more fusion tags from the GravyTrain collection, potentially fused to or replaced with other protein domains upon RF cloning. The cassette is integrated using pYM S1/S2 primers into a vacant genomic locus, such as auxotrophic mutant genes in common laboratory strains, e.g., w303 trp1−1. Alternatively, in cases where a gene knockout is desirable in addition to exogenous expression, S1/S2 primers for the knockout can be used to integrate expression cassette, achieving two genetic interventions in one step.

Notably, the protein sequence included in S1-S4-S2 cassettes can alternatively be used for S1/S4-mediated 5' tagging of endogenous genes (as above) rather than gene-independent integration. In addition, we introduce kanTS6 (e.g., construct pOS650), a new antibiotic selection marker as a replacement for kanMX4, wherein the expression of the resistance enzyme is terminated by the mild synthetic, physiologically inert promoter tSYNTH6 [23].

### GravyTrain Community allows shuffling between different construct types

**The GravyTrain community rationale.** As most constructs described above for yeast genetics include protein tags flanked by the universal RF hotspots GSGSGS_UP and GSGSGS_DOWN, these tags may be potentially shuffled using RF cloning by a single primer pair, regardless of the type of donor or target construct. We term this cloning framework **GravyTrain Community** (Fig 1D, S4A Fig). To demonstrate the applicability of this concept to research, we have created two GravyTrain Community sets of constructs as follows.

**GravyTrain Atg8 expression community constructs.** The first set was built for studying autophagy [26] by introducing different tags – mostly fluorescent proteins – as N-terminal fusions (replacement to 2xmCherry in a pre-existing expression construct) to Atg8 for genomic integration in yeast. Atg8 is a protein that is covalently conjugated to phosphatidylethanolamine (PE) on the surface of autophagic membranes, and is thus useful for live imaging of their biogenesis [27]. Resulting constructs are described (S2 Table, sheet "Community pFA6a-ATG8 constructs") and have a common standard layout (S4B Fig). Genomic integration is attained by amplifying the construct with a forward primer annealing at the ATG8 promoter and a reverse primer annealing at S2, both carrying homology to a genomic locus whose coded protein is to be eliminated – as in pFA6a-S1-S4-S2 exogenous expression described above. The tagged protein

Atg8 is consequently expressed under its native promoter and terminator alongside the endogenous untagged counterpart and is therefore not expected to interfere with the physiological conditions of wildtype autophagic membranes biogenesis. In addition to wildtype Atg8 (e.g., construct pOS222), we also include fluorescently-tagged previously-published mutations thereof, namely Atg8ΔR (e.g., construct pOS372) that does not require C-terminal processing by Atg4 [28] and is therefore useful to study the roles of Atg4 in autophagy [29]; Atg8$^{F104A,Y16A}$ (e.g., construct pOS532) for which *in vitro* membrane hemifusion activity was found to be severely impaired, and may therefore serve as identifier of artificially-stabilized, otherwise-transient hemifusion donor membranes *in vivo* [30]; and Atg8$^{P52A,R67A}$ (e.g., construct pOS693) whose interaction with autophagic cargo receptors and other Atg8-interacting proteins is impaired, and is therefore useful to study their respective roles in selective and nonselective autophagy [31].

**GravyTrain TECPR2 expression community constructs.** The second set of GravyTrain Community constructs was built by inserting C-terminal GravyTrain fusion tags following different truncation mutants of the mammalian multi-domain HSAN9-associated protein TECPR2 [32] in the ubiquitously employed mammalian transient overexpression construct pEGFP-C1. TECPR2 was recently implicated in neuronal dystrophy in a mice model of HSAN9 [33], and intracellularly in lysosomal targeting of autophagosomes – an activity attributed to its C-terminal TECPR domain [34]. Resulting constructs are described (S2 Table, sheet "Community pEGFP-C1 constructs") and have a common standard layout (S4C Fig). All TECPR2 sequences are followed by the non-GravyTrain TEVsite-3xFLAG-TwinStrepTag polypeptide to allow various affinity-based manipulations in addition to the functionality conferred by the GravyTrain tags. Upon expression in mammalian cell culture as previously reported for GravyTrain-deficient TECPR2 constructs [34], the added value of GravyTrain tags in this community construct set is expected to supply novel information contributing to our understanding of the differential contribution of each TECPR2 domain to the intracellular roles of the protein as a whole. For example, proteomic comparison of MiniTurboID-based proximity labeling [22] by the N-terminal domain WD domain (construct pOS607) and C-terminal TECPR domain (construct pOS601) may reveal distinct interaction partners that mediate the unique function of each. In addition to TEVsite-3xFLAG-TwinStrepTag-containing TECPR2 constructs, we also include here TECPR2-deficient pEGFP-C1 cassettes containing GravyTrain tags, either including TEVsite-3xFLAG-TwinStrepTag (e.g., construct pOS554) or excluding it (e.g., construct pOS709) – to allow characterization of other proteins in mammalian cell culture.

### Application of GravyTrain to studying autophagy in yeast

**Genetic background for nonselective autophagy.** To demonstrate the power of GravyTrain in molecular cell biology and yeast genetics, we devised and carried out the construction of a comprehensive library of yeast strains (S3 Table) for studying autophagy in yeast, using canonical constructs for genomic tagging, knockout, and exogenous expression knock-in, as well as community ATG8 constructs – described above. Most strains in the library overexpress *Oryza sativa* TIR1 by seamless genomic integration at the URA3 locus, in order to allow auxin-induced degradation of AID*-tagged proteins [35]. Furthermore, most strains (e.g., ORS477) are on the w303 background, which was reported to form large nonselective autophagic structures [36], which are here visualized by an additional, fluorescently tagged copy of Atg8 from the GravyTrain community ATG8 construct set. This copy is expressed by genomic integration instead of the ORF of Atg11 at the ATG11 locus to abolish Atg11-mediated selective autophagy and focus exclusively on nonselective autophagy [37]. Nevertheless, to study autophagy in more general conditions, some strains maintain Atg11 (e.g., ORS2664) or are derived from backgrounds other than w303 (e.g., ORS503).

**Characterization of Autophagic and Associated Proteins Using GravyTrain.** In addition to visualization of autophagic structures by fluorescently-tagged **Atg8**, combined with knockout of autophagy gene as negative controls or of other pathway genes study to study the effects of the latter of autophagy, the library includes tagging of various autophagy and non-autophagy proteins, e.g.,: **Sec61**-HaloTag7 (e.g., ORS1863) for ligand-mediate visualization [15] of the endoplasmic reticulum (ER) [38]; **Atg5**-MiniTurboID (e.g., ORS870) for biotinylation-based proximity studies [22] of Atg5;

Atg2-AID* (e.g., ORS514) for transient and quantitative auxin-induced depletion of Atg2; **Rps1b**-GBP (e.g., ORS2839) for GFP Binding Protein-mediated affinity towards avGFP derived proteins [39], including sfTq2 [14] fused to Atg8 (e.g., construct pOS471), thereby tethering Atg8 to the small ribosomal subunit protein Rps1b; **Atg24**-AaLS (e.g., ORS3102) for restricting the spatial distribution of Atg24 to the surface a lumazine synthase nanoparticle [21]; and knockin-mediated exogenous expression of mScarletI-tagged PX domain of Vam7 (e.g., ORS2066) for visualization of PI(3)P[23].

**Characterization of the Link Between Atg2 and Atg24 Using GravyTrain.** To illustrate the power of GravyTrain in the study of a molecular cell biology pathway *via* yeast genetics, we finally focus on two specific yeast strains recently employed in our characterization of the physical role played *in situ* by the Atg24-Atg20 complex. This complex binds autophagy-specific PI(3)P and local membrane curvature, thus stabilizing the rim of the nonselective autophagic isolation membrane under conditions of partial loss of Atg2-Atg18 complex activity [25]. The strain ORS3280 (Fig 1E, S5A Fig) (see experimental result in S5A Fig, "TMD_Sec71-Atg24-II-mScarletI Δvps38" [25]) contributed to this characterization was as follows: (i) w303 background allows visualization of large nonselective autophagic membranes upon treatment with the mechanistic target of the rapamycin (mTOR) inhibitor rapamycin [36] by standard widefield fluorescence microscopy. (ii) overexpression of seamlessly genomically-integrated TIR1 (not part of GravyTrain by itself) allows transient partial depletion of an AID*-fused protein (see below) by timely treatment with non-saturating concentration (10μM) of the naturally occurring auxin indole-3-acetic acid (IAA) [35]. (iii) autophagic membranes are visualized by an extra copy of **Atg8** fused to the relatively photostable and bright green fluorescent protein mNeonGreen [40] (Community ATG8 construct pOS222, hphNT1 marker), integrated instead of the ATG11 ORF to eliminate selective autophagy [37]. (iv) **Atg2** is tagged C-terminally with AID* (pFA6a-S3-S2 construct pOS343, tSYNTH8 termination, CgHIS3 marker), and its partial degradation leads to partial loss of rim constriction conferred by Atg2-Atg18 complex. (v) **Atg24** is tagged C-terminally with mScarletI for visualization (pFA6a-S3-S2 construct pOS249, tSYNTH8 termination, CgLEU2 marker). (vi) **Atg24** is artificially tethered to the cytosolic ER leaflet through N-terminal fusion of the transmembrane domain (TMD) of Sec71 (pFA6a-S1-S4 construct pOS553, pCUP1 promoter, natNT2 marker), leading to excessive opening of the rim. (vii) Atg24 rim opening activity is independent of the knocked out **Vps38** (pFA6a-S1-S2 construct pOS662), which recruits the core PI3-kinase complex to endosomes for endosomal PI(3)P generation [41]. The strain ORS3088 (S5B Fig) (see experimental result in Fig 2A, "Atg2-AID*" [25]) further contributed to validating the *bona fide* autophagic identity of abnormally-shaped autophagic membranes in conditions of partial Atg2-Atg18 complex activity. This was achieved by visualizing these structures as Atg8 and PI(3)P-positive projections from the vacuolar membrane towards the ER, as follows: (i-iv) as above for ORS3280. (v) The vacuolar membrane is visualized as expected at the base of autophagic structures by restriction enzyme-mediated genomic integration of a non-GravyTrain construct, overexpressing TagBFP-fused extra copy of the vacuolar membrane protein Pho8, with a LEU2 marker. (vi) The ER is visualized through fusion of **Sec61** to HaloTag7 [15] (construct pOS442, tSYNTH8 termination, CgHIS3 marker) by staining with the far-red Halo chemical ligand JF646 [42]. (vii) mScarletI-tagged PX domain of Vam7 (pFA6a-S1-S4-S2 construct pOS647, natNT2 marker, pCUP1 promoter, tSYNTH13 terminator), is integrated in the TRP1 locus to visualize PI(3)P.

**Conclusion: Studying Autophagy in Yeast Using GravyTrain.** In conclusion, the autophagy library of yeast strains in general and the described strains in particular facilitated the aforementioned experimental design in theory and scientific discovery in practice by combining the following features of GravyTrain: (a) Homologous recombination-mediated genomic transformations with amplified GravyTrain constructs. (b) Several selection markers including antibiotic resistance and auxotrophic markers. (c) Robust collection of protein tags. (d) Exogenous expression constructs. GravyTrain demonstrably has the power to facilitate scientific discovery through yeast genetics and molecular cell biology.

## Discussion

Scientific research is as powerful as the experimental tools that are employed for the unraveling of new phenotypes. Here we presented GravyTrain, a library of constructs that follow a unified design principle for protein characterization or

manipulation, gene knockout or knockin-mediated expression of exogenous genetic constructs. The canonical set of GravyTrain constructs for yeast genetics is compatible with the prevailing pYM amplification and homologous recombination scheme, thus providing yeast research groups with straightforward access to instrumentation of our library. The GravyTrain community framework allows shuffling of tags between different kinds of plasmid-borne genetic constructs other than the canonical pYM scheme using a single universal primer pair, as demonstrated here by construction of two GravyTrain community construct sets, namely ATG8 for visualizing autophagic structures in yeast, and pEGFP-C1 for protein characterization or manipulation of mammalian proteins by transient expression, particularly of the multi-domain protein TECPR2 and its structure-function truncation variants. GravyTrain Community thus has the potential to extend the power of GravyTrain's modular design and comprehensive versatility of protein fusion tags to research in model organisms other than yeast. Finally, we showed that a combination of different GravyTrain canonical and community constructs can be instrumented for the construction of a comprehensive library of yeast strains for molecular characterization of cellular pathways, exemplified here for nonselective autophagy in yeast.

While proving itself a powerful and easy toolbox for yeast genetics in molecular cell biology, GravyTrain maintains some inherent limitations nevertheless. Newly constructed plasmids must be cloned and sequenced prior to application to yeast genetics, tagged proteins lose their endogenous counterparts, amplification by PCR introduces mutations, and the number of successive genomic modifications is limited by the availability of selection markers and the introduction of off-target mutations in each transformation round. Future research should therefore focus on inventing novel and more sophisticated ways to introduce new genetic elements into the yeast genome while maintaining the functionality of endogenous proteins, while avoiding excessive duplication of genetic material in the process.

Like other recent additions to the experimental arsenal at the disposal of scientists, including SWAP-tag [5] and CRISPR/Cas9 [6], GravyTrain has the potential to advance in its own unique manner our understanding of biology. As GravyTrain was developed by standardizing and expanding on the previously published pYM construct library, we believe that future efforts may similarly extend the possibilities offered by GravyTrain, e.g., by augmentation with additional genetic elements and GravyTrain community construct types for models other than mammalian cell culture. As *de novo* design and construction of GravyTrain took form based on pre-existing simpler building blocks such as the pYM library, we suggest that future advancement in scientific methodology is not necessarily confined by the rate of technological breakthroughs but may rather be achieved by sheer creativity and liberty of invention. We call for broader adoption of our approach, which we term "GravyTrain", i.e., the acquisition of plenty through little effort [43].

## Materials and methods

### Constructs

GravyTrain genetics elements are listed and described (S1 Table), constructs are listed and described (S2 Table), and their sequence map ApE files are compressed in S1 File. All constructs were constructed by restriction free (RF) cloning [7] to substitute a DNA in a target construct with an external insert, using a mega-primer with regions of target construct homology flanking the insert. Mega-primers were either ordered as double-stranded gBlocks (IDT), or generated from template constructs, using pairs of oligonucleotides (Sigma) that anneal to template construct at their 3' and to the target construct at their 5'. pFA6a-S1-S4 and pFA6a-S1-S4-S2 constructs were derived from pYM-N2 [2], kindly gifted by Maya Schuldiner, by replacing the sequence between S1 and S4, and optionally adding the S2 sequence and preceding DNA immediately after S4. pFA6a-S3-S2 and pFA6a-S1-S2 constructs were derived from pYM25 [2], kindly gifted by Maya Schuldiner, by replacing the sequence between S1/S3 and S2. pFA6a-ATG8 constructs were derived from pFA6a-2xmCherry-Atg8-hphNT1, kindly gifted by Kuninori Suzuki. pEGFP-C1 constructs were derived from pEGFP-C1 [44]. Cassette of 3xmNG [45] was kindly gifted by Alejandro Colman-Lerner. Cassette of mScarletI [46] was kindly gifted by Gregory Finnigan. Cassette of 3vGFP and pCUP1(dim) [24] was kindly gifted by Morten Otto Alexander Sommer.

Plasmids other than GravyTrain which were used for strain construction include: pOS7, previously published as pNHK53 [35]; pOS148, originally published as pYM-N2 [2]; pOS202 (pUAS_F_E_C_CORE1-OsTIR1-tCaADH1), derived from published TIR2 [35] by RF-mediated promoter replacement into UAS_F_E_C_CORE1 [47]; pFA6a-kanMX4 was published [48]; and pFA6a-hphMX4, pFA6a-natMX4 were published [49]. Details on generation of specific constructs is available upon request.

### Yeast strains

Standard yeast maintenance, growth and transformation methods were used for generation of yeast strains – listed and described (S3 Table). Strains were generated using amplification from the indicated GravyTrain construct and genome-targeted homologous recombination as described [7], except for ORS475, ORS1150, and ORS1787, which were described [25]. Details on the generation of specific strains are available upon request.

### Supporting information

**S1 Table. Genetic Elements of GravyTrain.** The "Elements Overview" tab shows an overview of all element types and element names. The "Elements Details" tab includes a description, amino acid sequence and DNA sequence for each element. (XLSX)

**S2 Table. GravyTrain constructs.** The "Constructs Summary" tab includes a summary of all GravyTrain constructs, including a schematic representation of the included GravyTrain genetic elements. Each other tab lists GravyTrain constructs of a specific type described in the Results subsections. For each construct, a schematic textual description is listed, followed by GravyTrain genetic elements and optional type-specific additional genetic elements. (XLSX)

**S3 Table. GravyTrain yeast strains.**
For each strain, the parental strain, transformation cassette used to generate the strain, selection marker used in selecting for successful transformation, modified genomic gene, type of genetic modification, GravyTrain construct used for amplification of the transformation cassette, selection marker gene introduced in the genome by the transformation, promoter, ORF and terminator introduced, schematic representation of different GravyTrain element types introduced, ancestral lineage of the strain, full genotype, cumulative selection markers set used by the strain, and cumulative gene-modification set included in the strain are listed. (XLSX)

**S1 File. Maps of GravyTrain constructs.** ZIP file containing all construct map ApE files. Maps include the construct insert into the plasmid backbone – composed of consecutive GravyTrain genetic elements. For auxotrophic markers "N" is indicated instead of the marker sequence. (ZIP)

**S1 Fig. The modular design of GravyTrain allows restriction-free import, export, and shuffling of genetic elements.** (A) Common layout of GravyTrain constructs. Gray background – GravyTrain genetic elements. White background – a pair of GravyTrain restriction-free hotspots flank each genetic element. (B) Import of an element from an external construct into a GravyTrain construct. First, a forward primer consisting of hotspot A sequence followed by sense homology to the 5' end (green arrow) of the donated element ("D1"), and a reverse primer consisting of hotspot B reverse sequence followed by antisense homology to the 3' end (red arrow) of D1, are used to amplify the element from a non-GravyTrain donor plasmid into a mega-primer. Note that primer design is tailored for the specific element but is independent of target plasmid. Then, the mega-primer is used to amplify the target GravyTrain plasmid containing an old element 1 ("T1"). This generates a new GravyTrain plasmid, wherein T1 is substituted with the imported D1. (C) Export of a GravyTrain to an external construct. First,

a forward primer consisting of sense homology to the 5' half of the target insertion site (green region) followed by Hotspot A sequence, and a reverse primer consisting of antisense homology to the 3' half of the target insertion site (red region) followed by Hotspot B reverse sequence, are used to amplify the donated element 1 ("D1") from a GravyTrain donor plasmid into a mega-primer. Note that primer design is tailored for target plasmid but is independent of the specific element. Then, the mega-primer is used to amplify the target non-GravyTrain plasmid, thereby generating a new non-GravyTrain plasmid containing exported element. (D) Shuffling of a GravyTrain element between GravyTrain constructs. First, a forward primer consisting of Hotspot A sequence, and a reverse primer consisting of Hotspot B reverse sequence, are used to amplify the element D1 from a donor GravyTrain plasmid into a mega-primer. Note that primer design is independent of shuffled element and both donor and target plasmids. Then, the mega-primer is used to amplify the target GravyTrain plasmid, thereby generating a new GravyTrain plasmid, wherein the target element T1 is substituted by the donated element D1.
(TIF)

**S2 Fig. GravyTrain provides diverse and powerful fusion tags for protein characterization or manipulation.** Representative examples of protein fusion tags for characterization or manipulation of a Protein Of Interest (POI). Antibodies and other interactors for post-lysis characterization are indicated next to tags. (A) a flex for separating POI from another fusion tag; (B) TAP – including 6xHis, 6xFLAG and TwinStrepTag – for affinity-based biochemical purification and identification; (C) AID* for auxin-induced, transient and quantitative $SCF^{TIR1}$-mediated poly-ubiquitination and consequent degradation by the 26S proteasome; (D) ubc9ts for temperature-inflicted, complete degradation by the 26S proteasome; (E) superfolder mTurquoise2 (sfTq2) for constitutive cyan visualization; (F) HaloTag7 for transient, Halo chemical ligand-mediated visualization; (G) PMGpa2 for plasma membrane anchoring or- (H) PXVam7 for tethering to PI(3)P are fused to a fluorescent protein ("FP") for in situ visualization; (I) HA nanobody for tethering of POI1 to another, HA-tagged POI2; (J) SpyCatcher002 for covalent binding of POI1 to another, SpyTag002-fused POI2; (K) AaLS for spatial condensation on the surface of a multimeric nanoparticle. (L) TurboID for biotin proximity labeling.
(TIF)

**S3 Fig. Canonical GravyTrain constructs for genome editing in yeast.** Common layouts, in 5' to 3' sense orientation, of specific types of canonical GravyTrain constructs for genome editing in yeast, following homologous recombination-mediated integration in a genomic locus. Black background – genomic elements. White background – GravyTrain restriction-free hotspots. Gray background – GravyTrain genetic elements. Marker – selection marker for yeast transformation. Appendix – additional genetic element for integration, to be used in conjunction with construct (not in use in constructs provided herein). Prefix – polypeptide part, such as a linker or degron, to be expressed in fusion with the main functional tag and fused endogenous ORF. Promoter and Terminator – drive expression of the ORF. GSGSGS_UP and GSGSGS_DOWN – short neutral linkers flanking main functional tag. START and STOP – start and stop codons, respectively, for ORF translation. (A) Canonical pFA6a-S1-S2 constructs for gene knockout. Amplified by forward sense S1 and reverse antisense S2 primers and integrated between endogenous 5'UTR and 3'UTR instead of endogenous ORF, thereby knocking out its expression. (B) Canonical pFA6a-S3-S2 constructs for 3' gene tagging. Amplified by forward sense S3 and reverse antisense S2 primers and integrated between endogenous ORF and 3'UTR, thereby tagging the ORF at its C-terminus with the Prefix-Tag moiety and substituting endogenous terminator function with GravyTrain terminator within construct. (C) Canonical pFA6a-S1-S4 constructs for 5' gene tagging. Amplified by forward sense S1 and reverse antisense S4 primers and integrated between endogenous 5'UTR and ORF, thereby tagging the ORF at its N-terminus with the Prefix-Tag moiety and substituting endogenous promoter function with GravyTrain promoter within construct. (D) Canonical pFA6a-S1-S4-S2 constructs for knock-in of exogenous expression. Amplified by forward sense S1 and reverse antisense S2 primers and integrated between endogenous 5'UTR and 3'UTR instead of endogenous ORF, thereby knocking out its expression. Expression of exogenous Prefix-Tags polypeptide is driven by GravyTrain promoter and terminator within construct.
(TIF)

**S4 Fig. GravyTrain community constructs for broader applications of GravyTrain protein tags.** (A) Shuffling of GravyTrain protein fusion tags between different construct types: the universal GSGSGS_UP (red forward arrows) and GSGSGS_DOWN (green backward arrows) primer pairs are used to amplify mega-primers of GravyTrain tags (middle) from donor constructs of one type (left), and target them by another amplification to constructs from another type (pointed by diagonal arrows). This generates new constructs where the target tag is substituted by the tag from the donor tag. This shuffling capacity defines a community of GravyTrain constructs of different types – two examples of which follow below – beyond the sets of canonical yeast genetics constructs described above. (B) GravyTrain community pFA6a-ATG8 constructs allow visualization of autophagic membranes in yeast by expressing an extra copy of Atg8, fused N-terminally to a GravyTrain fluorescent protein ("FP") flanked by GSGSGS_UP and GSGSGS_DOWN. Construct is amplified by forward sense primer annealing at the beginning of the native ATG8 promoter ("pATG8") and a reverse antisense S2 primer, and genomically integrated between endogenous 5'UTR and 3'UTR instead of an endogenous ORF, thereby knocking out its expression. Included GravyTrain elements are described above. FP-Atg8 is expressed under control of native ATG8 promoter ("pATG8") and terminator ("tATG8"). (C) GravyTrain community pEGFP-C1 constructs allow characterization or manipulation of a transiently transfected overexpressed mammalian protein like a fragment of TECPR2 ("TECPR2(804-1407)"), fused C-terminally to a TEV cleavage site followed by 3xFLAG and TwinStrepTag affinity tags, and a variable GravyTrain protein fusion tag ("Tag") flanked by GSGSGS_UP and GSGSGS_DOWN. Upon transient transfection to a cell culture, protein and fused moieties are expressed as a single polypeptide under control of the CMV promoter ("pCMV") and SV40 terminator ("tSV40").
(TIF)

**S5 Fig. Instrumentation of GravyTrain for studying autophagy in yeast.** Presented are two genotypes of two strains – ORS3280 (B) and ORS3088 (C) – as examples for instrumentation of GravyTrain constructs in construction of a strain library for studying autophagy in yeast. (i-iv) are common to both strains (A): (i) w303 background for visualization of large nonselective autophagic membranes (not depicted); (ii) overexpression of non-GravyTrain TIR1 for AID*-mediate protein depletion (not depicted); (iii) integration of GravyTrain community ATG8 construct mNeonGreen-Atg8 (pOS222) for visualization of autophagic membranes, instead of the ORF of endogenous ATG11 to eliminate Atg11-mediated selective autophagy; (iv) C-terminal tagging of endogenous ATG2 ORF with GravyTrain construct of AID* (pOS343 for ORS3280, pOS525 for ORS3088) for transient partial auxin-mediated depletion of Atg2. (B) Strain ORS3280 was instrumented to establish that tethering the PI(3)P-binding protein Atg24 to the cytoplasmic ER membrane leaflet excessively opens the rim of the nonselective autophagic isolation membrane upon partial loss of Atg2-Atg18 complex activity, in a manner independent of endosomal PI(3)P: (v) C-terminal tagging of endogenous ATG24 ORF with GravyTrain construct of mScarletI (pOS249) for visualization of Atg24; (vi) N-terminal tagging of endogenous ATG24 ORF with GravyTrain construct of TMD$^{Sec71}$ (pOS553) for tethering of Atg24 to the cytoplasm leaflet of the ER membrane; (vii) knockout of VPS38 ORF for elimination of endosomal PI(3)P. (C) Strain ORS3088 was instrumented to establish the *bona fide* autophagic identity of aberrantly cup-shaped nonselective autophagic structures that form upon partial loss of Atg2-Atg18 complex activity, as Atg8, PI(3)P-positive projections from the vacuolar membrane towards the ER: (v) overexpression of non-GravyTrain TagBFP-Pho8, integrated at the LEU2 locus, to visualize the vacuolar membrane (not depicted); (vi) C-terminal fusion of endogenous ER protein Sec61 with GravyTrain construct of HaloTag7 (pOS442) for far-red visualization using Halo ligand JF646; (vii) expression of exogenous GravyTrain construct mScarletI-PX$^{Vam7}$ driven by copper-induced promoter pCUP1 and synthetic terminator tSYNTH13 (pOS647), by integration into the mutated TRP1 ORF for visualization of PI(3)P.
(TIF)

## Acknowledgments

Z.E. is the Harold Korda Chair of Biology incumbent and is supported by the Molly and Steven Elias Foundation and the Richard F. Goodman Yale/Weizmann Exchange Program.

The authors acknowledge Maya Schuldiner, Kuninori Suzuki, Alejandro Colman-Lerner, Gregory Finnigan and Morten Otto Alexander Sommer for their kind gifts. We also acknowledge Natalie Muskat, Milana Fraiberg and Damilola Isola for critical reading of this manuscript.

## Author contributions

**Conceptualization:** Oren Shatz, Zvulun Elazar.

**Data curation:** Oren Shatz.

**Funding acquisition:** Zvulun Elazar.

**Methodology:** Oren Shatz.

**Supervision:** Zvulun Elazar.

**Validation:** Oren Shatz.

**Writing – original draft:** Oren Shatz.

**Writing – review & editing:** Zvulun Elazar.

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
