## [Decision Letter · Decision Letter 0]

5 Jan 2025

Dear Dr. Elazar,

We look forward to receiving your revised manuscript.

Kind regards,

Mojtaba Kordrostami, Ph.D.

Academic Editor

PLOS ONE

Journal Requirements:

“The authors have declared that no competing interests exist.”

4. We are unable to open your Supporting Information file [S1_File.zip]. Please kindly revise as necessary and re-upload.

Reviewers' comments:

Reviewer's Responses to Questions

**Comments to the Author**

1. Is the manuscript technically sound, and do the data support the conclusions?

Reviewer #1: Yes

Reviewer #2: No

2. Has the statistical analysis been performed appropriately and rigorously?

Reviewer #1: N/A

Reviewer #2: No

3. Have the authors made all data underlying the findings in their manuscript fully available?

Reviewer #1: No

Reviewer #2: Yes

4. Is the manuscript presented in an intelligible fashion and written in standard English?

Reviewer #1: Yes

Reviewer #2: No

Reviewer #1: the abstract needs a little simplification as it is dense with more details and thus making it less overview in understanding to the scholarly community.

The simplification in this matter will be a better option so that it could be cool.

The figures 1-5 needs clear annotation and it could be done to make it clear.

A flowchart summarizing the workflow for GravyTrain construct generation and strain library development would enhance understanding.

Reviewer #2: Dear Editors,

I have reviewed the manuscript titled "The GravyTrain Toolbox for Molecular Cell Biology" (Manuscript PONE-D-24-45023). The manuscript presents a potentially transformative toolbox for genetic manipulation in yeast and other model organisms. However, I identified several areas that require significant revision to enhance the manuscript's clarity, validation, and generalizability.

Key areas of concern include:

The abstract is overly technical and does not clearly frame the problem GravyTrain addresses. The introduction lacks a strong comparison with alternative tools like CRISPR/Cas9 and SWAP-tag systems.

The manuscript lacks sufficient experimental evidence to substantiate the claimed advantages of GravyTrain, such as reduced recombination errors and broad applicability. Robust comparative data are necessary.

Several figures (e.g., Figure 3) are too dense and require simplification to enhance reader comprehension.

While yeast-related applications are well documented, the demonstration of versatility in mammalian systems is minimal and lacks experimental data.

The discussion overemphasized potential benefits without critically addressing limitations, scalability, or broader applications.

The concept of "GravyTrain Community" is intriguing but underexplored. Clearer plans for repository management and user contributions, as well as supplementary materials for reproducibility, are essential.

I have provided detailed suggestions in my review report to address these issues. I recommend major revisions, focusing on experimental validation, improved clarity, and expanded applications. With these revisions, the manuscript could become a highly valuable resource for molecular cell biology and yeast genetics.

Thank you for the opportunity to review this submission. Please let me know if further input is needed.

Sincerely,

**Do you want your identity to be public for this peer review?** For information about this choice, including consent withdrawal, please see our Privacy Policy

Reviewer #1: No

Reviewer #2: No

---

## [Author Response · Author response to Decision Letter 1]

15 Apr 2025

Please refer to our detailed response letter to the reviewers.

---

## [Decision Letter · Decision Letter 1]

30 Apr 2025

Dear Dr. Elazar,

We look forward to receiving your revised manuscript.

Kind regards,

Mojtaba Kordrostami, Ph.D.

Academic Editor

PLOS ONE

Journal Requirements:

Reviewers' comments:

Reviewer's Responses to Questions

**Comments to the Author**

Reviewer #1: All comments have been addressed

2. Is the manuscript technically sound, and do the data support the conclusions?

Reviewer #1: Yes

3. Has the statistical analysis been performed appropriately and rigorously?

Reviewer #1: Yes

4. Have the authors made all data underlying the findings in their manuscript fully available?

Reviewer #1: Yes

5. Is the manuscript presented in an intelligible fashion and written in standard English?

Reviewer #1: Yes

Reviewer #1: Although the work is thorough, there are instances where the methodology and results are arranged in a way that seems repetitious or overlaps. The experimental demonstration (autophagy strain library) and the explanation of the build designs should be more clearly separated for easier reading, perhaps with the use of subheadings or summary tables inside the body of the article.

The manuscript heavily relies on supplemental tables and figures to explain essential concepts. Integrating some of these key schematics or summaries into the main figures would make the manuscript more self-contained and reader-friendly.

**Do you want your identity to be public for this peer review?** For information about this choice, including consent withdrawal, please see our Privacy Policy

Reviewer #1: No

---

## [Author Response · Author response to Decision Letter 2]

7 Aug 2025

Journal Requirements:

We have now reviewed the list of references to ensure it is complete, correct and does not include any retracted papers.

We have uploaded our figure files to the PACE tool to ensure they meet PLOS requirements.

Reviewers' comments:

Reviewer's Responses to Questions

Comments to the Author

6. Review Comments to the Author

Reviewer #1: Although the work is thorough, there are instances where the methodology and results are arranged in a way that seems repetitious or overlaps. The experimental demonstration (autophagy strain library) and the explanation of the build designs should be more clearly separated for easier reading, perhaps with the use of subheadings or summary tables inside the body of the article.

As requested by the reviewer, we have now added subheadings to the section on studying autophagy in yeast using GravyTrain for easier reading.

The manuscript heavily relies on supplemental tables and figures to explain essential concepts. Integrating some of these key schematics or summaries into the main figures would make the manuscript more self-contained and reader-friendly.

We have now integrated some of the key supplementary information into the main figure.

---

## [Editor Report · Decision Letter 2]

31 Aug 2025

The GravyTrain toolbox for molecular cell biology

PONE-D-24-45023R2

Dear Dr. Elazar,

We’re pleased to inform you that your manuscript has been judged scientifically suitable for publication and will be formally accepted for publication once it meets all outstanding technical requirements.

Kind regards,

Mojtaba Kordrostami, Ph.D.

Academic Editor

PLOS ONE

Additional Editor Comments (optional):

All comments have been addressed.
---

## [Editor Report · Acceptance letter]

PONE-D-24-45023R2

PLOS ONE

Dear Dr. Elazar,

I'm pleased to inform you that your manuscript has been deemed suitable for publication in PLOS ONE. Congratulations! Your manuscript is now being handed over to our production team.

Kind regards,

on behalf of

Dr. Mojtaba Kordrostami

Academic Editor

PLOS ONE